# Six-Month Follow-Up of Periodontal Condition in Rheumatoid Arthritis and Ankylosing Spondylitis Arthritis Patients Undergoing Anti-Tumour Necrosis Factor-α Therapy

**DOI:** 10.3390/jcm13020448

**Published:** 2024-01-13

**Authors:** Ildikó Tar, Edit Végh, Renáta Martos, Boglárka Soós, Ildikó Márton, Zoltán Szekanecz

**Affiliations:** 1Department of Oral Medicine, Faculty of Dentistry, University of Debrecen, 4032 Debrecen, Hungary; tar.ildiko@dental.unideb.hu; 2Department of Rheumatology, Faculty of Medicine, University of Debrecen, 4032 Debrecen, Hungary; veghe22@gmail.com (E.V.); soos.boglarka@med.unideb.hu (B.S.); 3Department of Restorative Dentistry and Endodontics, Faculty of Dentistry, University of Debrecen, 4032 Debrecen, Hungary; martos.renata@dental.unideb.hu (R.M.); marton.ildiko@dental.unideb.hu (I.M.)

**Keywords:** rheumatoid arthritis, ankylosing spondylitis, periodontitis, anti-TNF treatment

## Abstract

In our present study, we aimed to assess the effects of anti-TNF therapy on periodontal condition in a mixed cohort of patients with rheumatoid arthritis (RA) and ankylosing spondylitis (AS). Moreover, we wished to determine whether the baseline dental condition of these patients would affect response to biological therapy. A cohort of 24 arthritis patients was consecutively recruited before starting anti-TNFα therapy. After the dropout of six patients, we evaluated the dental status of 18 subjects at baseline and after 6 months of biological therapy. Clinical responder (R) and non-responder (NR) status was determined after 6 months of anti-TNF treatment. Plaque index (PI), gingival index (GI), bleeding on probing (BOP), probing pocket depth (PPD), PPD_max_, clinical attachment loss (CAL), and CAL_max_ were determined. During the 6-month treatment period, six patients (3 RA and 3 AS) terminated the study prematurely as they did not respond to treatment (NR). Therefore, 18 patients were included in the full analysis. There were no major differences in PI, BOP, PPD, PPD _max_, CAL, and CAL_max_, among R and NR patients. TNF inhibition resulted in increased GI (0.65 ± 0.34 vs. 0.88 ± 0.30; *p* < 0.05), as well as decreased PPD_max_ (4 ± 1.94 vs. 2.72 ± 1.36; *p* < 0.05) and CAL_max_ (5.22 ± 2.56 vs. 2.72 ± 1.36; *p* < 0.05) after 6 months. Eight patients had incomplete canal fillings or dead pulps and/or apical periodontitis; six in the R and two in the NR group. In our present study, anti-TNF therapy seemed to worsen the extent of gingival inflammation (GI); however our results also do not support the reduction of mean CPD and CAL as reported by others.

## 1. Introduction

Tumor necrosis factor α (TNF-α) inhibitors have long been used for the treatment of rheumatoid arthritis (RA), ankylosing spondylitis (AS), as well as other inflammatory conditions [1,2,3]. RA has been connected with periodontitis mainly due to citrullination of α-enolase produced primarily by *Porphyromonas gingivalis*, but also by other periodontopathogenic bacteria [4,5,6]. Some studies also suggest that periodontitis may also be linked to AS [7,8], however this needs to be confirmed by larger studies. To our current understanding RA patients may have a higher risk for periodontitis compared to AS patients, possibly due to differences in disease pathogenesis. Yet, there have been only relatively few studies assessing the beneficial and potentially harmful effects of these biologics on the oral cavity, primarily on periodontal tissues [9,10,11,12]. Three studies had case-control design [9,10,11]. Among these studies one included psoriasis and systemic sclerosis in addition to RA patients [11]. The fourth longitudinal, interventional study evaluated the effects of 30 month adalimumab treatment in 20 RA patients [12]. In the case-control design studies (already anti-TNFα treated versus not treated), anti-TNF-α-treated subjects exerted less clinical attachment loss (CAL) [9,10,11,13]. In the follow-up study, periodontal indices, except plaque index, decreased significantly after treatment compared to baseline [12]. The effect exerted on periodontal tissues seems to change over time. At the beginning gingival inflammation starts to decline, while probing pocket depth (PPD) and clinical attachment loss (CAL) seem to be improved after 3 to 6 months [14].

In the 1970s periodontists rated the result of certain periodontal therapy (non-surgical—proper oral hygiene and scaling and root planning—or surgical—at that time flap surgeries) modalities. They wanted to evaluate the degrees of return of pocket depth and they related these observations to variations of periodontal attachment levels in the same locations, in order to assess the effectiveness of probing depth reduction [15,16]. Later regenerative surgical therapies were also evaluated the same way. Yet, this was not fully performed in the case of anti-TNF-α therapy.

The plaque induced periodontal disease that was demonstrated to be a risk factor for RA as periodontopathogenic bacteria was a predisposing factor for post-translational protein modification, citrullination catalysed by bacterial peptidylarginine deiminases (PPADs). Periodontal disease and chronic periapical periodontitis have fairly high overlap in bacterial flora and pathomechanisms, as chronic apical periodontitis is also associated with elevated levels of local and systemic proinflammatory mediators, while in tissue samples multiple forms of active immune and inflammatory reactions were demonstrated [17]. Chronic apical periodontitis is practically an inflammatory response against a polymicrobial infection of dental pulp origin. Chronic apical periodontitis can be presented histologically as dental granuloma or periradicular cyst. Even in that pathology the same defence and inhibitory mechanisms regulate the balance between bone resorption and regeneration causing the enlargement or shrinkage of periapical lesions. It has been demonstrated that citrullinated proteins and anti-citrullinated autoantibodies (ACPA) can be found in chronic apical periodontitis. This means that in chronic periapical periodontitis there are possible autoimmune pathways that can lead to the formation of autoantibodies, possibly leading to autoimmune diseases [18]. Clinically, the presence and frequency of this pathology has not yet been reported in RA or AS patients.

The effectiveness of any therapy is always an important question, and as a therapy is not fully conform to everybody, there could be different reactions. There could be patient-related, disease-related, treatment-related factors that affect therapeutical outcomes. The first two types even existed before the initiation of the therapy. Amongst patient-related factors the importance of gender, age, smoking, body mass index were demonstrated in terms of anti-TNF-α therapy. In the case of disease-related factors the importance of clinical behaviour, age at diagnosis, disease-duration, previous and concomitant treatments, laboratory values, e.g., CRP levels, and genetic predisposition and expression signature, cannot be denied. Disease severity has also an impact on therapeutical outcomes. In the case of RA it can be measured by disease activity score (DAS-28), while in the case of AS Bath ankylotic spondylitis disease activity score (BASDAI) is used. For therapy-related factors, there are two main time frames in assessing effectivity. Short term (12 weeks) and long term (over 3 months) evaluation after the start of the anti-TNF-α therapy present different problems [19]. Presumably not all effecting factors have already been revealed.

As the relationship between effects of anti-TNF therapy in RA and AS and periodontal condition is not fully understood, the aim of our present study was to conduct a longitudinal follow-up study on oral condition in a cohort of RA and AS patients undergoing anti-TNF-α therapy. We evaluated their oral mucosal, periodontal condition at baseline and after 6 months of biological therapy to determine the possible effects of anti-TNF therapy on dental condition. We also wished to demonstrate if these patients had any incomplete canal fillings or periapical lesions at baseline, and whether there was a clinically demonstrable change after 6 months on X-rays (periapicals and panoramic). Moreover, we wished to determine whether the baseline dental status was associated with anti-TNF treatment outcomes by comparing anti-TNF responders and non-responders.

## 2. Materials and Methods

### 2.1. Patients

A cohort of 24 arthritis (13 RA and 11 AS, determined by standard classification criteria) patients was consecutively recruited from the Department of Rheumatology, University of Debrecen. These patients were about to start on an anti-TNF-α (etanercept, ETN or certolizumab pegol, CZP) therapy. They were referred to the Department of Periodontology for oral screening before the initiation of biologics. After 6 months of biological treatment, they underwent check-up examinations applying the same investigations as at baseline. The baseline characteristics of the patients are included in Table 1.

Clinical responder (R) and non-responder (NR) status in RA was determined after 6 months of anti-TNF treatment by the EULAR response criteria originally described by Van Gestel et al. [20]. For AS we applied the ASAS 20 improvement criteria, developed and published by Anderson et al. [21].

The study was performed according to the Declaration of Helsinki and was approved by the National Ethical Board (ETT-KFEB, No. 3386-2011). All patients signed an informed consent.

### 2.2. Evaluation of Oral Status

First, a general clinical examination, including history taking and physical examination, was performed. The status and severity of periodontal disease were assessed by indices described below. Oral mucosa was examined clinically. If mucosal disorder was found, depending on its nature, histology was performed to underpin clinical diagnosis. In the case of verified diagnosis, patients obtained the appropriate treatment for their mucosal lesions. Those patients who were responders underwent a follow-up examination after 6 months of treatment. Post-treatment assessments were the same as those performed at baseline.

The evaluation of periodontal condition included full-mouth measurements using a periodontal probe. Plaque index (PI) [22] and gingival index (GI) [23] were determined. Bleeding on probing (BOP), probing pocket depth (PPD), and again its maximum (PPD_max_), the distance of marginal gingiva and cemento-enamel junction, as well as its maximum, were decided and added to matching PPD values by site, to produce clinical attachment loss and its maximum (CAL and CAL_max_) [24,25]. Panoramic X-rays and periapical radiographs were taken at baseline for all patients and after 6 months from responders. All these baseline indices were assessed separately in RA (*n* = 13) and AS (*n* = 11) patients to learn whether they can be handled as one group or not. The following step was to demonstrate whether R (*n* = 18) and NR (*n* = 6) patients might have a significantly different periodontal condition at baseline. Lastly, the improvement of periodontal condition in R patients was assessed by using the initial measurements and the differences of baseline and 6-month values of PPD, PPD_max_, CAL, CAL_max_.

All patients were also evaluated if they had any incomplete canal fillings and/or periapical lesions, at baseline, and R patients were re-evaluated after 6 months of anti-TNF-α therapy. Patients were offered dental treatment as they obtained a complete treatment plan for their problems. All of them wanted to go back to their original dentists, as, e.g., endodontic treatment can last longer and involves several seatings, and they do not live in the vicinity.

### 2.3. Statistical Analysis

Statistical analysis was performed using the SPSS 22.0 software (IBM, Armonk, NY, USA) as described previously [26,27]. Descriptive statistics for variables were performed. Means, standard deviation, and minimum and maximum of variables were calculated (Table 2 and Table 3). Baseline and 6 month data were compared to each other by paired-samples *t*-tests, while different group data were compared to each other with the help of two sample *t* tests (equity of means). Results were considered as significant if *p* values were ≤0.05. Relationship between baseline values (PPD_max_) and difference of baseline and 6-month PPD_max_ values was evaluated by Spearman rank correlation.

## 3. Results

As seen in Table 1, out of the 24 patients included in the study, 13 had RA and 11 had AS. There were 13 women and 11 men. Their mean age was 48.8 ± 13.8 years (range: 21–74 years) and their mean disease duration was 8.3 ± 6.2 years (range: 1–33 years). At baseline, before starting anti-TNF therapy, RA patients had a mean DAS28 of 4.88 ± 0.79, while AS patients exerted a mean BASDAI of 5.56 ± 1.22. At the beginning the two patient groups (RA and AS) were compared in terms of periodontal indices, including PI, GI, BOP, PPD, PPD max, CAL, CALmax, and periodontal disease stage. They were not different statistically in any of these parameters and their number seemed to be too low to deal with them as separate groups in statistics. As the examined parameters did not show statistical difference, these patients were unified into one group.

Altogether 20 patients (9 RA and 11 AS) started on 50 mg etanercept (ETN) SC weekly, while four patients (all RA) received certolizumab pegol (CZP; 400 mg at 0, 2, and 4 weeks and thereafter 200 mg every two weeks SC). During the 6 months of treatment period, six patients (3 RA and 3 AS) terminated the study prematurely as they did not respond to treatment (NR) according to the criteria described previously. Thus, they were not included in the final analysis, however, their baseline data were excluded for the R vs. NR comparison analysis. Therefore, altogether 18 patients (10 RA and 8 AS) completed the study and were included in the full analysis. There were no major differences in disease duration and periodontal data, including PI, BOP, PPD, PPD_max_, CAL, and CAL_max_, between R and NR patients (data is shown in Table 3) [28]. The baseline clinical and dental status of the six dropouts was different in age (R = 52.7 ± 11.8 years, NR = 37.2 ± 13.4 years, *p* < 0.05) and GI (R = 0.65 ± 0.39, NR = 0.52 ± 0.07, *p* < 0.05) from that of the others.

In the 18 patients, who completed the follow-ups, anti-TNF therapy significantly decreased disease activity in both RA and AS. In the 10 RA patients DAS28 decreased from 4.92 ± 0.87 at baseline to 3.22 ± 0.54 after 6 months (*p* = 0.01). Similarly, in the eight AS patients BASDAI decreased from 5.33 ± 1.18 at baseline to 1.97 ± 1.05 after 6 months (*p* < 0.01).

When comparing periodontal indices at baseline and after 6 months of anti-TNF treatment, GI increased (0.65 ± 0.34 vs. 0.88 ± 0.30; *p* < 0.05), while PPD_max_ (4 ± 1.94 vs. 2.72 ± 1.36; *p* < 0.05) and CAL_max_ decreased (5.22 ± 2.56 vs. 2.72 ± 1.36; *p* < 0.05) over time (Table 2). There were no significant changes in PI, BOP, PPD, and CAL over time (Table 1). We did not find any notable differences in periodontal indices at baseline or after 6 months between RA and AS patients.

Amongst the examined 24 patients eight had incomplete canal fillings or dead pulps (necrotized because of bacterial infection) and/or periapical lesions, called chronic apical periodontitis. In the R group there were six of them, while in NR group there were two cases present at baseline examination (periapical X-rays and panoramic X-ray). In the six R patients chronic periapical lesions could have been detected on X-rays, while amongst non-responders two patients had incomplete canal fillings with no periapical lesion present on X-rays. After 6 months with R patients it seemed that they had not gone through any dental interventions, not even at their own dentist. Periapical lesions were still detectable on X-rays, and they had not shown shrinkage at all.

By the end of the study period, oral lichen planus (OLP) lesions developed on the oral mucosa of two patients: one male and one female. In the male patient OLP appeared on the buccal mucosa and on the attached gingiva as desquamative gingivitis. In the female patient the lesions were only on the buccal mucosa. Severe forms did not appear later on. The clinical diagnosis was confirmed by histology in both cases. Clinically the lesions appeared as reticular (white, keratinized grid-like striae) and mild atrophic areas. The lesions were categorized as atrophic types located both on buccal mucosa and attached gingiva and they were followed up. Severe forms did not occur later on.

## 4. Discussion

RA and AS have long been treated with anti-TNF agents [1,2,3]. Moreover, RA [4] and maybe also AS [7,8] have been associated with chronic periodontitis. TNF-α inhibition suppresses inflammation, bone resorption, and thus joint damage in arthritides [1,2,3]. On the other hand, there have been very few studies on the possible effects of TNF-α inhibition on periodontitis [9,10,11,12]. Moreover, there have been no clinical studies on periapical lesions in terms of anti-TNFα therapy, although a study did examine the effect of adalimumab on the healing of apical periodontitis in ferrets [29], and they found beneficial effects. We also could not find a study on the role of baseline characteristics of periodontitis on the outcome of biologic treatment.

In our present study, anti-TNF therapy seemed to worsen the extent of gingival inflammation (GI) after 6 months, but it did not affect the location pattern of inflamed gingiva, as BOP did not change in a significant manner; however our results do not support the reduction of mean PPD and CAL as reported by others [9,12]. The patient cohort of Kobayashi et al. [12] differed from ours. Our results still suggest that biologic therapy might significantly reduce the maximum values of the latter two variables (PPD_max_, CAL_max_).

Partially similar findings were observed in studies conducted in the 1970s. In those early studies the higher the PPD (>3 mm) or/and CAL values were, the higher (>3 mm) the reduction of PPD and/or CAL gained upon periodontal treatments [15,16,30]. Even later on the same results were achieved [31]. What could be the reason for the behaviour of the gingival soft tissues? What happens in this case? Periodontitis is a combination of attachment loss (that results from the wandering of the junctional epithelium in the apical direction) and gingivitis, clinically defined as more than 10% of BOP around teeth. For the reduction of PPD and additionally CAL the pre-requirement is to reduce BOP percentage. The primary etiological factor is the accumulation of the dental biofilm finally leading to a Gramm-shift in the oral flora which represents more a dysbiosis than a true infection [32]; this was demonstrated clinically in periodontal disease already in the 60s. The histopathology of periodontal disease describes the certain stages of periodontal disease [33]. How it shifts parallel with the accumulation of the biofilm, starting with a homeostatic inflammation that reflects clinical health with low antigenic load of the dental biofilm, then advancing to mild gingivitis than severe gingivitis both reflecting chronic inflammation of the gingiva with extending volume (5–15% of the gingiva connective tissue). In the final stage the destruction of soft and hard tissues occurs in addition to gingival inflammation. In the gingiva all immunological processes are performed from recruitments of neutrophils, monocytes, and lymphocytes as in other parts of the body with the same cytokines and TNF-α. During the process of exerting gingival inflammation vasodilation, vascular proliferation in the dentogingival plexus occurs resulting in oedema of the marginal gingiva. In addition, the recruited cells perform a space creating process (degradation of collagens) to make enough space for themselves to be able to perform their duty in the area. Eventually, the consistency of the gingiva turns soft, it is increased in volume and there could be a mild atrophy of the gingival epithelium on the oral side. This is why BOP and GI can be measured because the gingiva turns softer than normal. The pressure provided by the periodontal probe must not exceed 25 g. If the gingiva volume grows, the PPD is going to increase as well. The CAL is made of PPD and the distance between the gingival margin and the cemento-enamel junction. If the PPD changes the CAL will also [33,34].

In the case of sulcular epithelium the cell attachments are loose originally, as the desmosomes do not hold very firmly between epithelial cells; in addition they are also not keratinized compared to the epithelium of the masticatory mucosa of the oral cavity, on the outer side. This is why neutrophils can step from the body to the crevice, and planktonic bacteria and the by-products of the biofilm can enter through these gates to the body. The successful elimination of the biofilm is impossible as the biofilm, and the exerted immune- and inflammatory answer are in different compartments. If the biofilm stays (bad oral hygiene) it provides a constant challenge for the local immune and inflammatory response. Practically this is why mechanical oral hygiene is so important. The constant stimulation by the dental biofilm keeps a constant stimulation of TNF-α levels in periodontal disease, resulting oedema, vasodilatation, recruitment of cells, and finally tissue destruction. The classical periodontal causative therapy aims for the reduction of bacterial numbers around periodontal tissues, thereby leading to the reduction of inflammation of the periodontium. The anti-TNF-α therapy aims for the immune and inflammatory reactions occurring in joints and as a side effect in periodontal tissues. Practically it is a symptomatic therapy in the case of periodontal tissues, but if there is a bad oral hygiene the stimulation of inflammation remains. As in the case of RA patients BOP is reduced in the first 3 months, although PPD and CAL decrease were demonstrated between 3–6 months of biologic treatment. Probably a change in the cell homestasis of the gingival inflammation changes. The established lesion has two main subtypes: B cell dominant or T cell dominant. Different T or B cell subsets seem to have different importance: T cell subsets like T regulatory, CD8^+^, and tissue resident γδ cells are important in the maintenance of the gingival homeostasis. However, in gingivitis the production of interleukin-17 and the secretion of osteoclastogenetic factors by activated T cells is crucial for the osteoclastogenesis by the initiation of RANKL pathways. The mucosa associated invariant T cells also express γIFN, TNF-α, and interleukin-17. This shows their possible role in disease initiation. The low levels of B memory cells also suggest a healthy periodontium [34,35]. So, probably parallel with the administration of anti-TNF-α therapy the composition of cells and the pattern of expressed mediators change causing the change in the BOP, GI, PPD, and CAL over time.

There is no study in humans on the effects of anti-TNFα therapy on periapical lesions (chronic apical periodontitis). Although hypercitrullination seems to be a significant pathological pathway in chronic apical periodontitis, *Porphyromonas endodontalis* does not possess PAD activity [18].

There has been no study that investigated whether baseline periodontal status could serve as a risk factor for the failure of biological therapy. Our study showed that younger age and lower baseline GI was significantly associated with therapeutic NR. Previous studies found that the custom periodontal treatment of RA and AS patients only improved RA patient disease activity but not AS patients. It is supposed that the interfaces with periodontal disease in etiological factors, immune and inflammatory answers are different in the two joint problems. In case of RA and periodontal disease TNF-α polymorphisms can play a significant role in both, in certain populations. Gingival inflammation and periodontal tissue destruction are regulated by similar cellular and molecular pathways of inflammation and tissue damage as in RA and AS [1,3,36,37]. TNF-α is involved in the pathogenesis of both arthritides and periodontitis [4,36,37]. Moreover, the periodontopathogenic bacterium *P. gingivalis* has also been implicated in RA [4,5]. The endotoxin (LPS) of this bacterium is a potent activator of macrophage TNFα production. Moreover, its PAD enzyme citrullinates bacterial α-enolase resulting in autoantibody production against this enzyme [4,5]. ACPA can also activate macrophages to produce TNFα [4,38,39]. Therefore, in younger patients with lower levels of periodontal inflammation, the production of TNF-α might be less pronounced. Other mediators, such as interleukin-1β (IL-1β), IL-6, granulocyte-macrophage colony stimulating factor (GM-CSF), vascular endothelial growth factor (VEGF), and prostaglandin E_2_ production also showed significant correlation with disease activity [40]. In arthritis patients with less gingival inflammation indicated by lower GI, TNF-α is probably not the major inducer of either periodontitis or RA. This may be one explanation, why anti-TNF treatment is not so successful in younger patients with lower GI. In addition, there is indeed a subset of RA patients with less systemic inflammation, where TNF inhibition may not be so effective and anti-TNF therapy can be discontinued without inducing flare [41]. At younger age additional health problems that also increase the level of TNF-α are less frequent. Mostly severe periodontal disease (just gene polymorphisms at the background) starts around the age of 35. So, the clinical appearance of the disease needs time.

We postulate that younger age and mild gingival inflammation in such patients may result in anti-TNF therapy failure. Certainly, further studies are needed to elucidate this issue.

Our study has certain strengths and limitations. This is one of the very few studies where the effects of biologics on the course of periodontitis over time has been investigated. This is the first study to investigate the frequency and type of chronic periapical periodontitis in RA and AS patients clinically. This might be the very first study linking baseline periodontal status with therapeutic responses. Moreover, we used a wide array of indices to assess periodontal condition. The possible limitations include the relatively low number of patients. There were only six NRs, which made the statistical analysis difficult. We also did not perform a power analysis, and we did not have a control group; we only performed a longitudinal follow-up so the baseline was the basis for comparison. Finally, the relatively low patient number did not permit comparison of RA and AS.

Yet, we think this is an original contribution as we were able to assess the longitudinal effects of anti-TNF on periodontal indices and could associate baseline periodontal status with therapeutic responses. Further analyses in larger cohorts might be needed in order to further elucidate the relationships between periodontal status and targeted therapies in patients with inflammatory rheumatic diseases.

## Figures and Tables

**Table 1 jcm-13-00448-t001:** Patient characteristics.

	RA	AS	Total
*n*	13	11	24
female:male	10:3	3:8	13:11
age (mean ± SD) (range), years	55.9 ± 9.8 (35–74)	43.6 ± 12.4 (21–62)	48.8 ± 13.8 (21–74)
disease duration (mean ± SD) (range), years	9.1 ± 8.3 (1-–33)	7.2 ± 7.0 (1–26)	8.3 ± 6.2 (1–33)
RF positivity, *n* (%)	10 (77)	-	-
ACPA positivity, *n* (%)	11 (85)	-	-
DAS28 (baseline) (mean ± SD)	4.88 ± 0.79	-	-
BASDAI (baseline) (mean ± SD)	-	5.56 ± 1.22	-
Treatment (ETN, CZP)	9 ETN, 4 CZP	11 ETN	20 ETN, 4 CZP

Abbreviations: ACPA, anti-citrullinated protein antibody; AS, ankylosing spondylitis; BASDAI, Bath Ankylosing Spondylitis Disease Activity Index; CZP, certolizumab pegol; DAS28, 28-joint disease activity score; ETN, etanercept; RA, rheumatoid arthritis; RF, rheumatoid factor; SD, standard deviation.

**Table 2 jcm-13-00448-t002:** Baseline and post-treatment periodontal indices in patients completing the study (*n* = 18).

	Mean ± SD	Significance
PI-0 *	0.61 ± 0.61	n.s.
PI-6 *	0.52 ± 0.63	
GI-0	0.65 ± 0.34	*p* < 0.05
GI-6	0.88 ± 0.30	
BOP-0	0.06 ± 0.23	n.s.
BOP-6	0.19 ± 0.13	
CPD-0	1.60 ± 0.74	n.s.
CPD-6	1.49 ± 0.48	
CPD_max_-0	4.00 ± 1.94	*p* < 0.05
CPD_max_-6	2.72 ± 1.36	
CAL-0	2.14 ± 1.98	n.s.
CAL-6	2.10 ± 1.28	
CAL_max_-0	5.22 ± 2.56	*p* < 0.05
CAL_max_-6	2.72 ± 1.36	

* 0 and 6 indicate baseline and 6-month post-treatment values, respectively. Abbreviations: BOP, bleeding on probing; CAL, clinical attachment loss; CAL_max_, maximum CAL; CPD, clinical probing depth; CPD_max_, maximum CPD; GI, gingival index; n.s., non-significant; PI, plaque index.

**Table 3 jcm-13-00448-t003:** Effect of baseline periodontal indices on response to anti-TNF therapy *.

	Mean ± SD	Significance
age-R (years)	52.7 ± 11.8	*p* < 0.05
age-NR (years)	37.2 ± 13.4	
PI1-R	0.61 ± 0.68	n.s.
PI1-NR	0.29 ± 0.15	
GI1-R	0.65 ± 0.39	*p* < 0.05
GI1-NR	0.52 ± 0.07	
BOP1-R	0.06 ± 0.23	n.s.
BOP1-NR	0.00 ± 0.00	
CPD1-R	1.60 ± 0.74	n.s.
CPD1-NR	1.28 ± 0.39	
CPD1_max_-R	4.00 ± 1.94	n.s.
CPD1_max_-NR	2.83 ± 1.83	
CAL1-R	2.14 ± 2.14	n.s.
CAL1-NR	1.37 ± 1.37	
CAL1_max_-R	5.22 ± 2.46	n.s.
CAL1_max_-NR	3.17 ± 2.40	

* The analysis included 18 responders (R) and 6 non-responders (NR). Abbreviations: BOP, bleeding on probing; CAL, clinical attachment loss; CAL_max_, maximum CAL; CPD, clinical probing depth; CPD_max_, maximum CPD; GI, gingival index; n.s., non-significant; PI, plaque index.

## Data Availability

Data is contained within the article.

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
