# Peer review of "Six-Month Follow-Up of Periodontal Condition in Rheumatoid Arthritis and Ankylosing Spondylitis Arthritis Patients Undergoing Anti-Tumour Necrosis Factor-α Therapy"

_jcm, 2024, doi:10.3390/jcm13020448_

Round 1

Reviewer 1 Report

Comments and Suggestions for Authors

Review of : Six-Month Follow-Up of Periodontal Condition and Oral 2 Mucosa Lesions in Arthritis Patients Undergoing Anti-Tumour 3 Necrosis Factor-α Therapy

The study by Tar et al is of interest but lacks specifically clinical information. The most important research question was to determine whether baseline dental condition would affect the outcome of biological therapy. This question is not answered. Also the patient groups are not discussed separately.

Major comments:

-          What was the clinical and periodontal status of 6 non responders who dropped out?

-          The authors give no information on what the criteria are for responder and non-responder, they are  determined by an not recent reference, and these are different for RA and AS. Please describe the criteria.

-          Why did non responders not get a follow up examination?

-          A Table 1 with general characteristics is missing: age, gender, disease duration autoantibody profile, treatment, etc. Now there are 2 places in the manuscript were age and gender information is given which are different, line 67-68, line 117-118. This is however not enough.

-          Because RA and AS are different diseases the patients cannot be put together in one table but the periodontal data should be given separately for each patient group.

-          Tables should be arranged differently

-          Line 125: criteria above: these criteria are not given, what was the improvement in DAS28 or BASDAI?

-          Line 177: a reference is given here in full.

-          Very old references are used.

-          The discussion is not coherent

o   the non responders are completely left out of the further analysis

o   the groups are put together and not discussed separately

Comments on the Quality of English Language

English language is ok

Author Response

The study by Tar et al is of interest but lacks specifically clinical information. The most important research question was to determine whether baseline dental condition would affect the outcome of biological therapy. This question is not answered. Also the patient groups are not discussed separately.

We thank the reviewer for the helpful comments. We made all the necessary corrections in RED colour. The effect of baseline dental condition on TNFi responses was indeed investigated, see Table 3 and the text. We compared R and NR patients. Also we only had 24 patients in total, 13 AS and 11 RA. This relatively small number did not make separate analysis possible. This information is now included in the text on page 7.

Major comments:

What was the clinical and periodontal status of 6 non responders who dropped out?

The baseline clinical and dental status (except GI) of the 6 dropouts was not different from that of the others (data not shown). This info is now included on page 7.

The authors give no information on what the criteria are for responder and non-responder, they are  determined by an not recent reference, and these are different for RA and AS. Please describe the criteria.

Even if it seems to be an old publication, the response criteria in RA first published by van Gestel et al is still in use, so we used this in RA. The reviewer is right, response in RA and AS are different. In AS we used the ASAS 20 improvement criteria developed by Anderson et al, now added to the text and cited (page 5).

Why did non responders not get a follow up examination?

Due to non-response they did not want to further participate in the study.

A Table 1 with general characteristics is missing: age, gender, disease duration autoantibody profile, treatment, etc. Now there are 2 places in the manuscript were age and gender information is given which are different, line 67-68, line 117-118. This is however not enough.

A Table 1 with patient characteristics is now added and cited. We renumbered the previous tables 1 and 2 to 2 and 3, respectively. We added age, gender, disease activity, autoantibody profile, treatment info.

Because RA and AS are different diseases the patients cannot be put together in one table but the periodontal data should be given separately for each patient group.

As discussed above due to the relatively small number of patients we could not assess the two groups separately. We understand that RA and AS are two different diaseases, however, we used standard odontological methods, which can be equally applied to both diseases.  

Tables should be arranged differently

We changed table arrangement as requested. Now mean and SD are expressed in one cell.

 Line 125: criteria above: these criteria are not given, what was the improvement in DAS28 or BASDAI?

Now we added a section on efficacy results by adding DAS28 and BASDAI changes in the 18 patients undergoing follow-ups. In the 18 patients, who completed the follow-ups, anti-TNF therapy significantly decreased disease activity in both RA and AS. In the 10 RA patients DAS28 decreased from 4.92±0.87 at baseline to 3.22±0.54 after 6 months (p=0.01). Similarly, in the 8 AS patients BASDAI decreased from 5.33±1.18 at baseline to 1.97±1.05 after 6 months (p<0.01).  

 Line 177: a reference is given here in full.

We are not sure what the reviewer means here, however, we now added a reference from our previous studies on the Statistical analysis.

Very old references are used.

Now we added some new references (see in RED) plus we expanded the text as requested.

The discussion is not coherent

We now reworded the Discussion. We significantly extended it with new information

Reviewer 2 Report

Comments and Suggestions for Authors

Although the subject is fairly interesting, this is not an original study (from what I could see in the manuscript).

The abstract is very long for a fairly basic research.

The introduction is not well structured.

The study design is very poor.

The discussions section is severely lacking in relevant comparisons with the literature.

Limitations (and there are many...) are only mildly touched upon.

All in all the manuscript needs intense work.

Please see the enclosed PDF for further details on other pressing issues suggestions for improvement.

Comments on the Quality of English Language

English is fine

Author Response

Although the subject is fairly interesting, this is not an original study (from what I could see in the manuscript).

We thank the reviewer for assessing our manuscript. There have not been many studies ont he effects of biologics on dental conditions or the effects of detailed baseline dental status on biologic responses. Thus, we think our study is fairly original. We made necessary changes in ther manuscript in RED colour.

The abstract is very long for a fairly basic research.

We now shortened the abstract to below 250 words which is standard in most papers.

The introduction is not well structured.

We now restructured the Introduction and added new information and new referemces.

The study design is very poor.

We now reworded the last paragraph of the Introduction to clarify our study design. Further details of the disign have been added tot h Methods section.,

The discussions section is severely lacking in relevant comparisons with the literature.

We now extensively expanded the Discussion with further comparisons with the literature and additional references.

Limitations (and there are many…) are only mildly touched upon.

We now expanded the limitations as requested.

Round 2

Reviewer 1 Report

Comments and Suggestions for Authors

The manuscript has improved greatly by adding more data and improving the discussion. The authors have addressed al the raised questions.

Author Response

We thank Reviewer 1 for accepting our revision and for the positive comment.

Reviewer 2 Report

Comments and Suggestions for Authors

There were many more suggestions in the included pdf in the previous review, the authors did not even look at them. Please address them ALL. 

Comments on the Quality of English Language

English is fine

Author Response

We thank the reviewer for the additional comments. Yes, we admit that we did not realize that there was also a PDF attachment, therefore we addressed only the listed issues. Now we address all issues attached as stickers to the manuscript. However, as we worked on the original word file before for revision, we continue to do this on the revised word file rather than the edited file. We made further changes in the manuscript in RED colour, and we reply to the comments on the margin of the manuscript.

Round 3

Reviewer 2 Report

Comments and Suggestions for Authors

The  manuscript has been improved